

# Spatial changes in zooplankton communities in a strong human-mediated river ecosystem

Robert Czerniawski[1,*] and  Monika Kowalska-Góralska[2,*]

[1] Department of General Zoology, Centre of Molecular Biology and Biotechnology, University of Szczecin, Szczecin, Poland
[2] Institute of Biology, Department of Hydrobiology and Aquaculture, Wrocław University of Environmental and Life Sciences, Wroclaw, Poland
[*] These authors contributed equally to this work.

## ABSTRACT

River damming causes a decrease in water current velocity which leads to an increase in richness and abundance of organisms atypical for running waters. Zooplankton is a representative example of such organisms. The influx of zooplankton from carp ponds is an additional factor that increases richness and abundance of zooplankton in rivers. We hypothesized that zooplankton dispersing from the carp ponds colonize the impoundments in river and the richness of zooplankton increase in impoundments by development of new species, not observed in the upstream. The zooplankton was collected monthly from April to September of 2013 and 2014. Sampling sites were located in the Barycz river (in the lotic sections and in the dam impoundments), as well as in its tributaries, which are the outlets of carp ponds. The most changes in zooplankton richness and abundance were observed at sites located within the dam impoundments, especially in relation to the lower values of the current velocity. Since the abundance of pelagic rotifers, cladocerans and copepods in the carp pond outlets was similar to that at lower sites in the Barycz, the influence of the carp pond outlets on the abundance in the dam and lotic sections was significant. The river itself in its impounded sections provides advantageous conditions for retention and colonization by a high abundance of zooplankton dispersing from the carp ponds, and for the development of species not occurred in the upstream, which, in turn, increases richness.

# INTRODUCTION

Catchment management is one of the factors that determines the functioning of a region and comprises numerous technical measures that depend on a region's needs, river-bed maintenance, agricultural drainage, water impoundments for fish farming or securing energy needs, flood control systems, and water retention (*Jones et al., 2004*; *Soranno et al., 2015*; *Ullah, Jiang & Wang, 2018*). On the other hand, there are also works, such as restoration of river stretches, which intend to make the area more appealing, increase its economic and biodiversity potential, as well as the environmental value (*Dufour & Piégay,*

Corresponding author
Monika Kowalska-Góralska,
monika.kowalska-
goralska@upwr.edu.pl

*2009*; *Zawal et al., 2016*). Consequently, all the technical measures, which alter the river-bed and the hydrological conditions, affect the biological functioning of the lotic sections.

Human-made dams are one of the most important technical measures that break the river continuum. Oftentimes, they cause irreversible alterations in rivers, and distort their natural flow (*Allan & Castillo, 2007*). Such alterations are reflected in physicochemical and biological variables, as well as the loss and replacement of typical lotic species by typical stagnant water species. A large number of studies examine how large dams in large rivers cause changes in environmental conditions, whereas few papers address the impact of small dams in small rivers or streams. However, small dams impounding a water of up to 4 m cause similar changes in river ecosystems, albeit on a much smaller scale. In small rivers, even small dams can affect the flora and fauna composition, by inducing rapid changes in hydrological and physicochemical variables (*Cumming, 2004*; *Wu et al., 2010*; *Zhou et al., 2008*). Small dams create distinct physical and ecological conditions that are relatively different from the ones found in free-flowing lotic sections. This is manifested by a reduction of current velocity, increase of water retention time and increase of nutrient content.

These new conditions e.g., long water retention time and low current velocity are suitable for the growth of zooplankton populations, that in turn are a good indicator of hydrological changes (*Czerniawski & Domagala, 2014*; *Ot'ahel'ová & Valachovič, 2002*; *Zhou et al., 2008*). Zooplankton can develop in stagnant waters or in slow-flowing waters with current velocities lower than 0.1 m s$^{-1}$ (*Czerniawski & Domagala, 2014*; *Zhou et al., 2008*). Given that, zooplankton reacts rapidly to the hydrological changes caused by dams. Until now, the influence of dams on zooplankton communities has been studied primarily in large rivers, with dammed water up to more than 15 m (*Akopian, Garnier & Pourriot, 1999*; *Doi et al., 2008*; *Pourriot, Rougier & Miquelis, 1997*; *Żurek & Dumnicka, 1989*). However, only a few studies have investigated the influence of small dams on zooplankton communities (*Czerniawski & Domagala, 2014*; *Zhou et al., 2008*). Small dams promote the formation of biotopes of the ecotone type, in which the zooplankton community develops features similar to or different from those communities in lakes and large reservoirs (*Czerniawski & Domagala, 2014*; *Zhou et al., 2008*).

Another process leading to alterations in a man-changed environment is catchment transformation, i.e., creation of retention reservoirs or fish ponds joined with main channel of the river. The amount of dead and live organic matter (including zooplankton) significantly increases in such reservoirs (*Kloskowski, 2011*; *Meijer et al., 1990*; *Rahman et al., 2008*). An example of artificial reservoirs are carp ponds, which lead to the increase in organic matter, and cause an influx of the organic matter to the rivers. High biomass of common carp cause an increase of nutrients, turbidity and suspended solids, hence, these basins have a strong influence on water quality and aquatic community structure (*Nieoczym & Kloskowski, 2014*; *Parkos, Santucci & Wahl, 2003*). The dispersion of carp-pond plankters can enrich the river with high densities or new zooplankton species. The inorganic and organic nutrients in the water column can drift to the outlet section thereafter, and then be dispersed into rivers.

None of the abovementioned studies investigated the concurrent impact of zooplankton influx from carp ponds and small dams impounding water of these carp ponds on the zooplankton communities in rivers. Moreover, in the case of the present study, small river dams can create suitable conditions for the dispersed carp-pond plankters, and for the development of new zooplankton species (not present in the upstream). That is why the sum of alterations occurring in a small river and its immediate surroundings can enhance changes in qualitative and quantitative structure of zooplankton in a different manner than dams alone. This phenomenon has not been examined. Therefore, we decided to investigate the concurrent impact of both factors on zooplankton communities in a small river: (i) impact of the dams impounding the river water, and (ii) impact of the zooplankton influx from the carp ponds outlets.

Construction of a small dam and the zooplankton influx from the carp pond outlets would significantly shape the zooplankton community in a river environment. The overall goal of the study was to examine the distribution of the zooplankton richness and abundance in a river dammed in many places and connected with the carp ponds. The following tasks were proposed: a comparison of the zooplankton communities between the free-flowing lotic sections and the impounded sections of the river (1), and a comparison of the zooplankton communities between the carp pond outlets and the river (2). We hypothesized that zooplankton dispersing form the carp ponds can colonize the impoundments in river (1) and the richness of zooplankton increase in impoundments by development of new species, not observed in the upstream (2).

## METHODS

### Study area

This study was performed in the Barycz and its few tributaries which are outlets of carp ponds (drainage of the Oder river, SW Poland) (Fig. 1). The Barycz is a river in the Lower Silesian Province in southwestern Poland. It is a right tributary of the Oder river. The Barycz is 139 km long, has a catchment area of 5,526 km$^2$, and is characterized by a mean slope of 0.035%. The catchment of the Barycz Valley Landscape Park is an important wetland reserve. Carp ponds in the Barycz Valley are the largest carp breeding ponds in Central Europe. They constitute the biggest complex of fish ponds and the oldest group of artificial reservoirs in Poland; they were built as early as the 15th and 16th century for carp farming (Gąbka, Dolata & Antonowicz, 2007). The location of the ponds in the Barycz Valley allows the inflow of gravitational water. The Barycz provides water for the carp ponds, therefore numerous dams and impoundments are located along the river. Currently, the total area of fish ponds in the catchment of the Barycz covers approximately 7,500 ha. The proportion of the agricultural area of the Barycz in the catchment is approximately 80%. From August to October, an accelerated discharge of water from the carp ponds into the Barycz is observed. In the examined stretch of the Barycz 31 dams with a hydraulic height of 1.2 m–4.6 m are located.

Sampling sites were located in the Barycz, in the lotic, free-flowing sections (R1, R2, R3, R4), at the dam impoundments (D1, D2, D3, D4, D5), and in the tributaries of the Barycz,
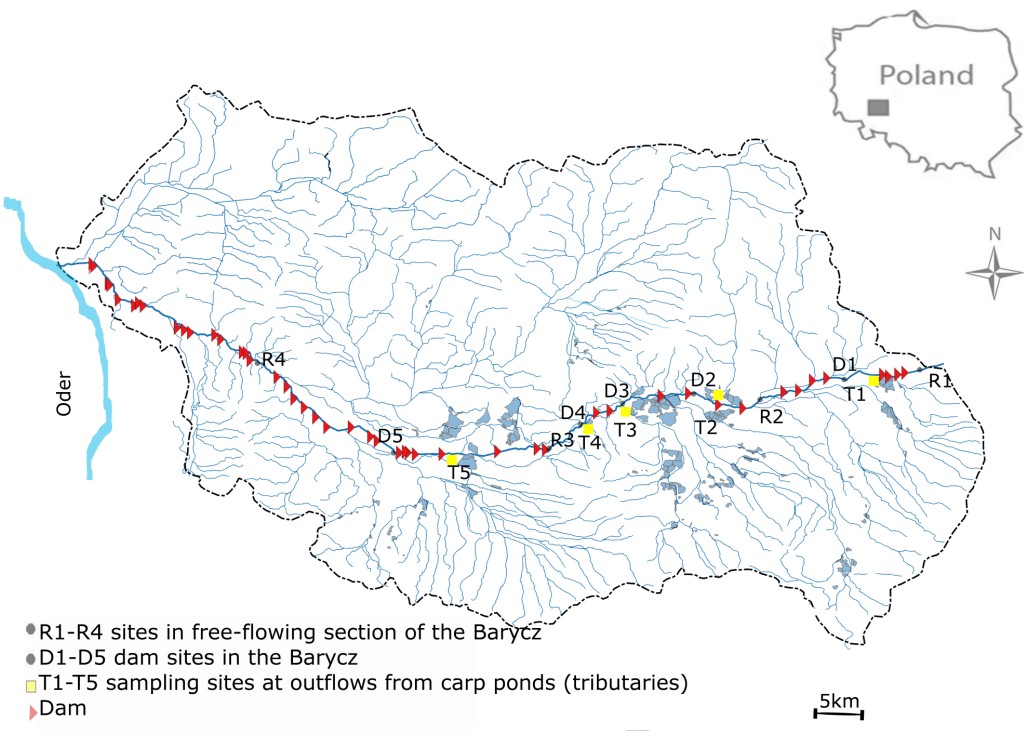

Poland

N

Oder

R4

D3    D2    D1    R1
D4    T3    T1
R3 T4    T2    R2
T5

D5

- R1-R4 sites in free-flowing section of the Barycz
- D1-D5 dam sites in the Barycz
- T1-T5 sampling sites at outflows from carp ponds (tributaries)
- Dam

5km

**Figure 1    Study area.**

which are also carp pond outlets (T1, T2, T3, T4, T5) (Fig. 1). The hydraulic heights of the dams at D1, D2, D3, D4 and D5 were respectively, 2.5 m, 2.5 m, 1.6 m, 2.3 m and 1.6 m. In Barycz river Lagrangian scheme according to which cross-sections were sampled in a downstream sequence with the sampling interval approximating the time of travel between sites. The sampling sites of Barycz were chosen under account (1) influence of the carp pond outlets, (2) influence of the dam impoundments on zooplankton communities in downstream, (3) easy access.

Sand was the dominant component in the sediments of the lotic sections of the Barycz. Whereas, the sediments of the impoundments were marshy (contained leaf litter), with many small floodplains and slackwater areas. The riparian zone of the impoundments was densely covered by emerging macrophytes (dominants: *Glyceria maxima, Phragmintes communis, Sparganiun erectum*), whereas the central zone of the bed impoundment was covered by elodeids (dominants: *Potamogeton natans, Ceratophyllum demersum, Potamogeton perfoliatus, Sagittaria sagittifolia*). The riparian zone of the lotic sections was sporadically covered by emerging macrophytes (dominants: *Glyceria maxima, Sparganiun erectum*), whereas its central bed was sporadically covered by *Potamogeton perfoliatus* and *Sagittaria sagittifolia.*

Tributaries—outflows of the carp ponds—were regulated channels with a sandy bottom. The owners of carp ponds remove regularly aquatic vegetation from both the riparian zone and the central part of the bed to ensure free and unobstructed water flow to the Barycz.

Tributaries between the carp ponds and the Barycz did not include any slack-water areas or floodplains. Therefore, the morphological conditions of the outlets were characterized by a relatively fast water current (from 0.36 m s$^{-1}$ to 0.48 m s$^{-1}$). No particular field sampling permits with regard to the site locations, e.g., the national park or other protected area and protected species, were needed.

## Sampling methods

The zooplankton was collected monthly from April to September of 2013 and 2014 ($n = 12$). At each site, 50 L of water was collected from the river current. The samples were collected using a Van Dorn 5-liter water sampler (KC Denmark) at five depths: 20%, 40%, 60%, 80%, and at the surface (*Czerniawski & Domagala, 2014*). At each depth level, 10 L of water was collected to obtain 50 L of water. The water was filtered through a plankton net with a mesh of 30 μm. The samples were then concentrated to 150 ml and fixed in a 4–5% formalin solution. The contents of the samples were counted in a Sedgewick-Rafter counting chamber in ten 3-ml sub-samples. A Nikon Eclipse 50i microscope was used for identification. Species were identified using the keys described by *Nogrady, Wallace & Snell (1993)* and *Rybak & Błędzki (2010)*.

Measurements of chlorophyll *a* content were made *in situ* using the Hydrolab DS 5 multiparameter probe (OTT Hydromet, Loveland, CO, USA). At each site, water velocity, width and depth were measured with an electromagnetic water flow sensor OTT (Kempten, Germany) to determine water discharge. A cross-section of the stream channel was divided into five vertical subsections. In each subsection, the area was obtained by measuring the width and depth of the subsection, and water velocity was determined using a current meter. Water discharge in each subsection was calculated by multiplying the subsection area by the measured velocity. The total discharge was then counted by summing up the discharge values of each subsection.

## Data analyses

Rotifers were divided into two categories, according to habitat preference–pelagic species (plankton) and benthic, epiphytic, epilithic species associated with the substratum otherwise known as benthic species (*Radwan, 2004*; *Zhou et al., 2008*). Taxonomical similarity between the sites was calculated using the Sorensen's index. We used the Kruskal–Wallis test ($P < 0.05$) for checking the significance of differences in the chlorophyll *a* content, discharge and current velocity values, and the richness (number of species) and the abundance of each zooplankton group among the sites. Post-hoc multiple comparisons of mean ranks for all groups were conducted ($P < 0.05$) to determine the significant differences in the zooplankton richness and abundance between the sites. Cluster analysis based on Euclidean distance was used to identify groups of similar sites of tributaries, impoundments and lotic sections with regard to the richness and the abundance of zooplankton. To illustrate the similarities between the sites (lotic sections R1–R4 *vs.* dammed sections D1–D5 *vs.* tributaries T1–T5) in terms of all taxa abundance, nonmetric multidimensional scaling ordination (nMDS) technique was used. The grouping in the nMDS ordination was based on the Bray–Curtis distances (*Oksanen et al., 2016*). In order

to determine the influence of environmental factors on richness and the abundance of zooplankton, Spearman's correlation was applied ($P < 0.05$). For the evaluation of the correlation between environmental factors *versus* the zooplankton richness and abundance in the Barycz, the following factors were considered: ND—number of dams above the site in the Barycz, DCP—distance between the site in the Barycz and the closest carp pond in the pond system, NCO—number of the carp pond outlets above the site in the Barycz, TPelRotR—richness of pelagic rotifers in the closest tributary above a site in the Barycz, TBenRotR—richness of benthic rotifers in the closest tributary above a site in the Barycz, TClaR—richness of cladocerans in the closest tributary above a site in the Barycz, TCopR—richness of copepods in the closest tributary above a site in the Barycz, TPelRotA—abundance of pelagic rotifers in the closest tributary above the site in the Barycz, TBenRotA—abundance of benthic rotifers in the closest tributary above the site in the Barycz, TClaA—abundance of cladocerans in the closest tributary above the site in the Barycz, TNaupCA—abundance of copepods Nauplii in the closest tributary above the site in the Barycz, TCopA—abundance of copepods in the closest tributary above the site in the Barycz.

The environmental conditions at the tributary sites were very similar and had absolutely no effect on the development of the zooplankton composition. The zooplankton composition in the tributaries (carp pond outlets) depended entirely on the conditions present in the carp pond upstream. Hence, when calculating the correlation between zooplankton and environmental factors, the environmental conditions at the outlets were not taken into consideration, as they could alter the end results.

## RESULTS

### Environmental factors

The means of the measured abiotic variables of chlorophyll *a* content, discharge, current velocity and values of constant factors are shown in Table 1. Generally, a spatial increase of the chlorophyll *a* values was observed. However, in the lotic sections, the chlorophyll *a* content decreased in relation with the dam sites. Significantly lower current velocity values were observed at the dam sites than at the lotic section. From R2, a significant increase in discharge occurred, and another rapid increase in discharge was observed from D5.

### Taxonomic composition

A spatial increase in the number of zooplankton taxa was observed (Table 2). A higher number of taxa was noted at the dam sites in the Barycz than in its tributaries (carp pond outlets) (Table 2). Only at R1—the first site of the Barycz (not influenced by the tributaries and the dams)—the number of taxa was the lowest.

All taxa observed in tributaries were also detected in the Barycz (Table 2). However, not all species observed in the Barycz, especially the ones at dam sites, occurred in samples collected at the tributaries. Among them were rotifers: *Scaridium longicauda, Synchaeta oblonga, Testudinella. patina, Trichocerca. capucina, Trichocerca similis*; cladocerans: *Eurycercus glacialis, Bosmina. gibbera, Ceriodaphnia laticaudata, Disparolona rostrata, Graptoleberis testudinaria, Pleuroxus. trigonellus, Simocephalus lusaticus*; copepods:

**Table 1** Mean values ± SD of chlorophyll a content, discharge and current velocity values and constant factors of the examined sites in the Barycz river.

| Site | Chl *a* (μg l⁻¹) | Discharge (m³ s⁻¹) | Velocity (m s⁻¹) | ND | DCP (km) | NCO |
|------|------------------|--------------------|--------------------|----|----------|-----|
| R1 | 3.7 ± 2.3 a | 0.27 ± 0.12 a | 0.27 ± 0.06 a | 0 | – | – |
| D1 | 8.3 ± 5.4 ab | 0.34 ± 0.15 b | 0.08 ± 0.01 b | 5 | 9.7 | 1 |
| R2 | 10.2 ± 6.6 ab | 2.00 ± 0.87 c | 0.23 ± 0.05 a | 8 | 15 | 1 |
| D2 | 13.5 ± 7.8 bc | 2.16 ± 0.94 cd | 0.06 ± 0.02 b | 11 | 0.3 | 2 |
| D3 | 16.4 ± 9.2 bc | 2.38 ± 1.04 cde | 0.06 ± 0.02 b | 13 | 0.1 | 3 |
| D4 | 18.2 ± 8.9 bc | 2.55 ± 1.11 cde | 0.05 ± 0.02 b | 15 | 0.1 | 4 |
| R3 | 11.6 ± 6.9 bc | 2.63 ± .1.15 cde | 0.16 ± 0.04 a | 16 | 7.3 | 4 |
| D5 | 22.9 ± 10.3 c | 4.27 ± 1.86 de | 0.02 ± 0.02 b | 23 | 6.9 | 5 |
| R4 | 14.7 ± 6.6 bc | 5.77 ± 2.51 e | 0.39 ± 0.09 a | 31 | 36 | 5 |

**Notes.**

Chl *a*, chlorophyll *a* content; velocity, current velocity; ND, number of dams above the site in the Barycz river; DCP, distance between the site in the Barycz river and the closest carp pond in the pond system; NCO, number of carp pond outlets (T) above the site in the Barycz river.

Letters in column show significant differences between the sites.

R1–3, sites in the free-flowing waters of the Barycz river; D1–5, sites in the dams of the Barycz river; T1–5, sites in the tributaries of the Barycz river.

*Mesocyclops gigas.* The frequency (number of samples in which a species occurred in relation to number of all samples expressed as a percentage) of different taxa was higher at the Barycz sites than in the tributaries e.g., *Brachionus* sp, *Cephalodella apocolea, Euchlanis dilatata, Mytilina* sp*., Alona* sp*., Eucyslops macruroides, Eudiaptopus gracilis* (Table 2).

## Sorensen's similarity

The Sorensen taxonomic similarity index value between the tributaries and the sites below them in the Barycz was more than 60% (Fig. 2). The farther the distance from the tributary to the Barycz sites, the lower the Sorensen similarity index value. Taxonomic similarity between the subsequent sites of the Barycz was relatively high (more than 60%). The highest similarity (more than 90%) was observed between the last sites in the Barycz—D5 and R4.

## Richness of zooplankton

In the case of pelagic rotifers, the higher the mean richness in the tributaries, the higher values of this parameter were observed at the subsequent sites below a tributary in the Barycz (Fig. 3). However, no significant differences in richness of pelagic rotifers occurred between these sites ($P > 0.05$). The mean richness of pelagic rotifers in R1 was significantly lower than in each tributary ($P < 0.05$), apart from T1 ($P > 0.05$) (Table 3). The mean richness of benthic rotifers was similar at each site and did not differ significantly between the sites in the tributaries, dams and lotic sections of the Barycz ($P > 0.05$) (Table 3).

In the case of cladocerans, mean richness differed significantly from most sites in the Barycz and its tributaries only at R1 ($P < 0.05$) (Table 3), where the mean richness was lower than compared with the other sites. The mean richness of copepods differed significantly only between R1 and T2. Contrary to pelagic rotifers, crustaceans (cladocerans and copepods) were characterized by a slightly higher mean richness at the dam sites than

**Table 2  Taxonomical composition and frequency of total zooplankton at each site of the Barycz River and its tributaries.** R1–4 sites on free-flowing stretch of the Barycz, D1–5 sites on the dams, T1–5 sites on the tributaries. Values in table are related to frequency.

| Taxa | R1 | T1 | D1 | R2 | T2 | D2 | T3 | D3 | T4 | D4 | R3 | T5 | D5 | R4 |
|---|---|---|---|---|---|---|---|---|---|---|---|---|---|---|
| *Ascomorpha saltans* | | | | | | | 14 | 14 | 14 | 14 | | | | |
| *Asplanchna priodonta* | | 14 | 14 | 14 | 43 | 43 | 14 | | 14 | | 43 | 71 | 43 | 29 |
| Bdelloidea | *36* | 100 | ***86*** | 86 | 100 | 57 | 57 | 71 | 86 | 64 | 71 | 86 | 64 | 64 |
| *Brachionus angularis* | 14 | 14 | 14 | | 14 | | 21 | 14 | 14 | | 36 | | 29 | 14 |
| *Brachionus calicyflorus* | | 29 | | | 29 | 43 | 43 | 29 | 29 | 29 | 29 | 29 | 29 | 21 |
| *Brachionus diversicornis* | | | | | | | 29 | 29 | 29 | 14 | 14 | 29 | 14 | |
| *Brachionus falcatus* | | | | | | | 14 | | 14 | 29 | 14 | | | |
| *Brachionus leydigii* | | | | | 14 | | | | | 14 | | | | |
| *Brachionus quadridentatus* | 14 | 14 | 7 | 14 | | | 29 | 14 | 14 | 7 | | 14 | | |
| *Bracionus rubens* | | 14 | 14 | | | | | | | | | 14 | 14 | 14 |
| *Brachionus urceoralis* | | | 7 | 14 | | | | | 14 | | | 14 | 14 | 14 |
| *Cephalodella apocolea* | 21 | | 21 | | | 14 | | 21 | | | 7 | 14 | 7 | |
| *Colurella adriatica* | | 14 | 14 | | | | 7 | | | | 7 | | | |
| *Euchlanis dilatata* | | 7 | | | | 21 | | 7 | | 7 | 14 | | 7 | 29 |
| *Filinia longiseta* | | 14 | | 14 | 43 | 14 | 57 | 14 | 43 | 21 | | 43 | 14 | 14 |
| *Kellicottia longispina* | | 7 | | | 14 | | 14 | | 57 | 14 | | 57 | 14 | 14 |
| *Keratella cochlearis* | 21 | ***100*** | 71 | 14 | *100* | 43 | 86 | 29 | 100 | 29 | 71 | 100 | 57 | 71 |
| *Keratella quadrata* | 14 | *71* | 36 | ***43*** | 71 | 43 | *100* | 29 | 86 | 29 | 50 | ***86*** | **57** | **64** |
| *Lecane clocterocerca* | | 14 | | | | 14 | 29 | | | | | 14 | | |
| *Lepadella acuminata* | 7 | | | 14 | | | | | 14 | | | | | |
| *Lepadella ovalis* | | | | | | 14 | | | 29 | 14 | | | | |
| *Mytilina crassipes* | | | | 14 | 14 | 14 | | | | | | | 14 | 14 |
| *Mytilina mucronata* | 14 | 14 | *21* | 29 | 29 | 43 | | 43 | 29 | 36 | 36 | 29 | 21 | 57 |
| *Platyias quadricornis* | 14 | | | 14 | | 14 | *21* | | 29 | 14 | | | | |
| *Polyarthra longiremis* | | 14 | 14 | 14 | | | 14 | | | | | | | |
| *Polyarthra vulgaris* | | | | 14 | | | 14 | 14 | 29 | 14 | 29 | 57 | 14 | 14 |
| *Scardinium longicaudatum* | | | | | | 14 | | | | | | | | |
| *Synchaeta pectinata* | | 14 | | | | | 14 | 14 | 14 | | | | | |
| *Synchaeta oblonga* | | | | | | | | | | 14 | | | | |
| *Testudinella patina* | 7 | | 7 | 7 | | | 7 | | | | 14 | | 14 | 7 |
| *Trichocerca capucina* | 14 | | | | | | | | | | | | | |
| *Trichocerca similis* | | | 7 | | | | 14 | | 14 | 14 | | | 14 | 14 |
| *Trichotria pocillum* | | | 14 | | | | 14 | | | | | | | |
| *Acropeus harpae* | 7 | | | | | | 14 | 14 | 29 | | 14 | | 14 | 14 |
| *Alona costata* | 7 | | | | | | 14 | 14 | | 14 | | | | |
| *Alona guttata* | | | | | | 29 | 14 | 14 | | 14 | | 14 | | |
| *Alona quadrangularis* | | | | | | | | | | | | 14 | 14 | 21 |
| *Alona rectangula* | | 14 | | 14 | 14 | 14 | | 29 | 14 | 29 | 14 | | 14 | 7 |
| *Alonella nana* | | | | | | | 14 | 14 | 29 | 14 | 29 | | 21 | 29 |
| *Eubosmina coregoni* | | 14 | 14 | 14 | 43 | 14 | 14 | 14 | 14 | 14 | 14 | | 7 | |
| *Eubosmina longicornis* | 7 | 21 | | 7 | | | 14 | 14 | 29 | 21 | 29 | 43 | 29 | 7 |

**Table 2** (*continued*)

| Taxa | R1 | T1 | D1 | R2 | T2 | D2 | T3 | D3 | T4 | D4 | R3 | T5 | D5 | R4 |
|---|---|---|---|---|---|---|---|---|---|---|---|---|---|---|
| *Eurycercus glacialis* | | | | | | 14 | | | | 14 | | | | |
| *Bosmina gibbera* | | | | | | | | | | | 14 | | | |
| *Bosmina longirostris* | *14* | **71** | *50* | **43** | **86** | **71** | **86** | **43** | **100** | **71** | *57* | **86** | *64* | *86* |
| *Ceriodaphnia quadrangula* | | 14 | | 14 | 43 | 29 | 43 | 43 | 43 | 29 | 43 | 43 | 14 | 7 |
| *Ceriodaphnia laticaudata* | | | 14 | | | | | | | | | | 14 | |
| *Chydorus ovalis* | | 29 | 29 | 14 | | 29 | 29 | 29 | | 29 | 29 | | 29 | 7 |
| *Chydorus sphaericus* | *21* | **93** | *64* | *50* | **71** | *64* | **100** | *50* | **100** | *43* | *50* | **79** | *71* | *79* |
| *Daphnia cucullata* | | | 14 | | 29 | 29 | 29 | 7 | 29 | 14 | | 29 | 14 | |
| *Daphnia longispina* | | | | | 29 | 14 | | 14 | | 14 | | 57 | 29 | 7 |
| *Diaphanosoma brachyurum* | | | | | | | 14 | 14 | | 14 | 14 | | | |
| *Disparolona rostrata* | | | | | | | | | | | 14 | | | |
| *Graptoleberis testudinaria* | | | | | | | | 14 | | 29 | | | | |
| *Peracantha truncata* | | 14 | 14 | | | | | | | | | | | |
| *Pleuroxus trigonellus* | | | | | | 14 | | | | | | | 14 | 14 |
| *Scapholeberis mucronata* | | | | 14 | 29 | 14 | 14 | 14 | | | 14 | | 14 | 14 |
| *Simocephalus lusaticus* | | | | | | | | 14 | 7 | | | | | |
| *Simocephalus serrulatus* | | | | | | 14 | 14 | 14 | 14 | 14 | 10 | 14 | 14 | 14 |
| *Simocephalus vetulus* | | | | | | 14 | 14 | | 14 | | 14 | | | |
| Nauplii Cyclopoida | **79** | **100** | **79** | **79** | **100** | **93** | **100** | **79** | **93** | **79** | 86 | **100** | 93 | ***100*** |
| Copepodit Cyclopoida | 21 | 86 | 43 | 71 | **100** | 79 | 100 | 86 | 100 | 64 | 71 | 100 | 57 | 57 |
| *Acanthocyclops robustus* | 7 | 71 | 43 | 29 | 79 | 64 | 71 | 43 | 36 | 29 | 29 | 50 | 29 | 29 |
| *Eucyclops macruroides* | 7 | | 14 | | 14 | 21 | | 36 | 21 | 29 | | 29 | 29 | 7 |
| *Eucyclops serrulatus* | | 14 | | | 14 | 14 | 14 | | 7 | 7 | | 7 | | |
| *Eudiaptomus gracilis* | | | | | | | 7 | | 7 | 21 | 14 | | 14 | 7 |
| *Megacyclops gigas* | | | | 14 | | | | 7 | | 14 | | | 14 | |
| Taxa number | 20 | 27 | 25 | 27 | 25 | 34 | 35 | 42 | 35 | 44 | 33 | 31 | 40 | 34 |

**Notes.**
**bold** > 20%; **italic-bold** 10–20%; *italic* > 5% of the mean total abundance of zooplankton at the site of each stream; taxa that were lesser than 5% of the mean total abundance are not marked.

in the tributaries ($P > 0.05$). In the lotic sections of the Barycz, no significant decrease in the richness in relation with the impoundments was observed. Therefore, the mean richness of crustaceans was similar between the Barycz (the dam and the lotic section sites) and its tributaries.

The richness similarity dendrogram of cluster analysis shows that the tributary sites are rather separated from other sites with respect to pelagic rotifers, cladocerans and copepods (Fig. 4). The richness of these groups at the first lotic section sites is farther from the last sites in the impoundments. The dendrogram does not show such differences in the richness of benthic rotifers.

## Abundance of zooplankton

Pelagic rotifers contributed to the increase in zooplankton abundance in the Barycz below the tributaries or dams (Fig. 5). However, no significant differences in the abundance of pelagic rotifers occurred between these sites ($P > 0.05$) (Table 4). The mean abundance of pelagic rotifers at R1 was significantly lower than at the tributary sites ($P < 0.05$).

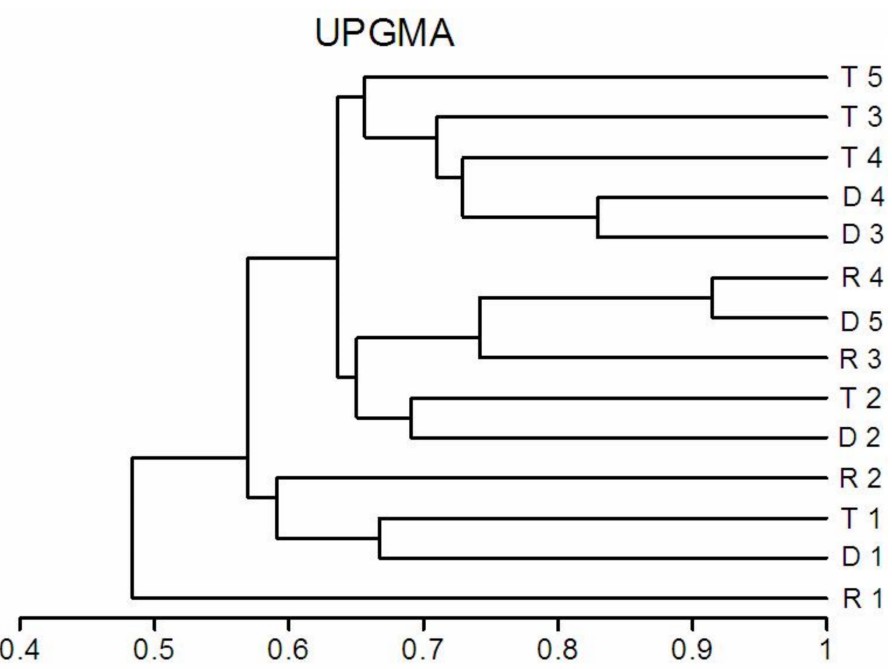

**Figure 2** Sorensen's similarity index value between the free-flowing (lotic) sites of the Barycz river (R), the dam sites of the Barycz (D) and the tributaries of the Barycz (T).

**Table 3** All significant differences (*P*-values) between the studied sites for the richness of the zoo-plankton groups (post-hoc multiple comparisons of mean ranks for all groups).

| Group | Site | T1 | T2 | D2 | T3 | T4 | R3 | T5 | D5 | R4 |
|---|---|---|---|---|---|---|---|---|---|---|
| Pelagic Rotifera | R1 | | * | | *** | *** | | *** | | |
| | D1 | | | | * | * | | ** | | |
| | R2 | | | | ** | ** | | *** | | |
| | D2 | | | | | | | *** | | |
| | D3 | | | | | | | * | | |
| | D4 | | | | | | | * | | |
| Cladocera | R1 | * | *** | * | *** | *** | * | *** | * | * |
| Copepoda | R1 | | * | | | | | | | |

**Notes.**
*$P < 0.05$.
**$P < 0.01$.
***$P < 0.001$.

The mean abundance of benthic rotifers did not differ significantly between the sites ($P > 0.05$) (Table 4). Generally, with respect to the subsequent sites, a higher abundance of benthic rotifers occurred in the Barycz (dam and lotic sections) than in the tributaries (Fig. 5).

Similarly to pelagic rotifers, cladocerans and copepods (Nauplii and the remaining stages). demonstrated a increase in the mean abundance in the Barycz below the tributaries (Fig. 5). However, contrary to the richness values, the abundance of crustaceans in the dam

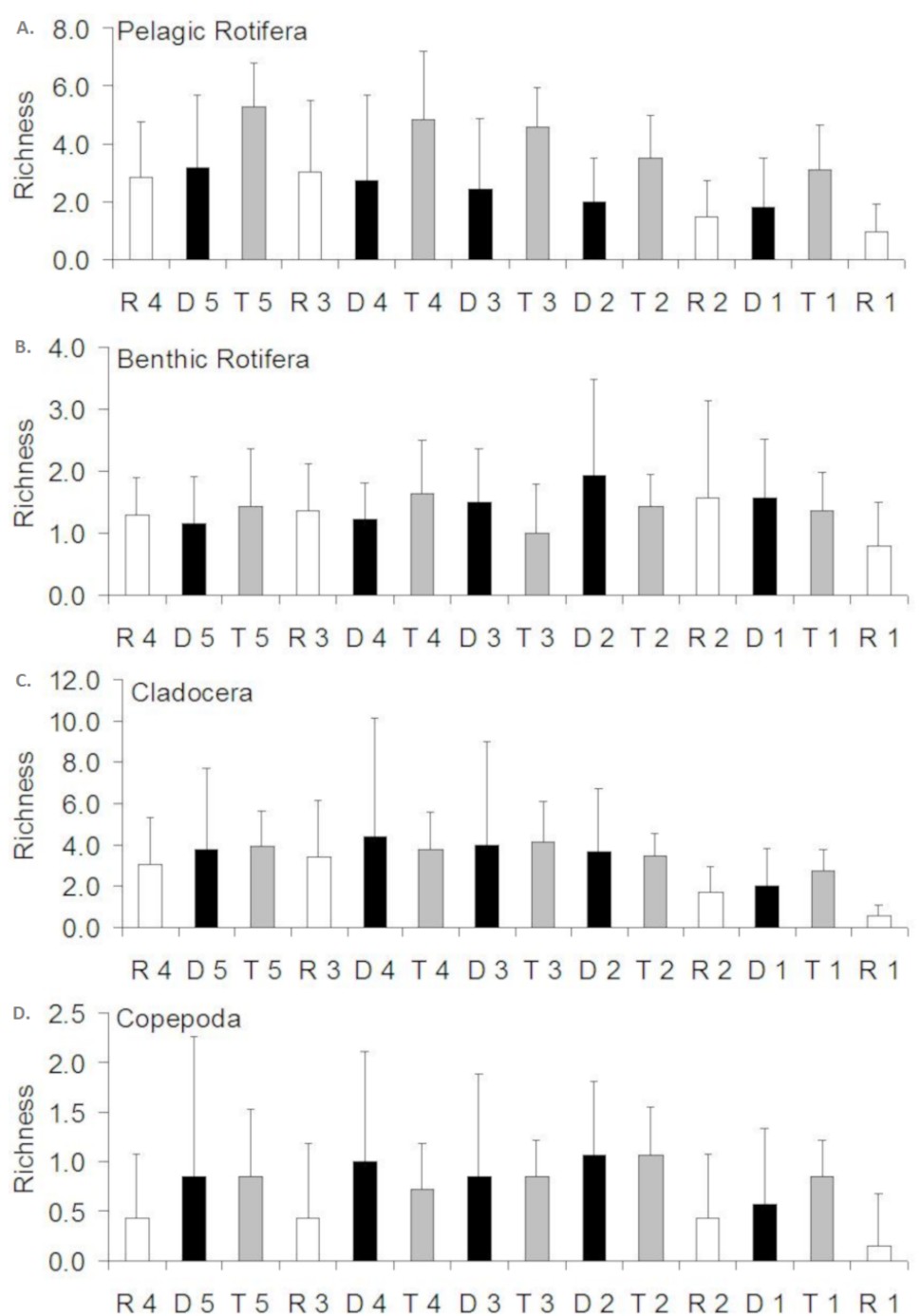

**Figure 3 Spatial distribution of the zooplankton richness in the Barycz river.** R1–4, sites in the free-flowing waters of the Barycz; D1–5, sites in the dams of Barycz; T1–5, sites in the tributaries of the Barycz. (A) Pelagic Rotifera, (B) Benthic Rotifera, (C) Cladocera, (D) Copepoda.

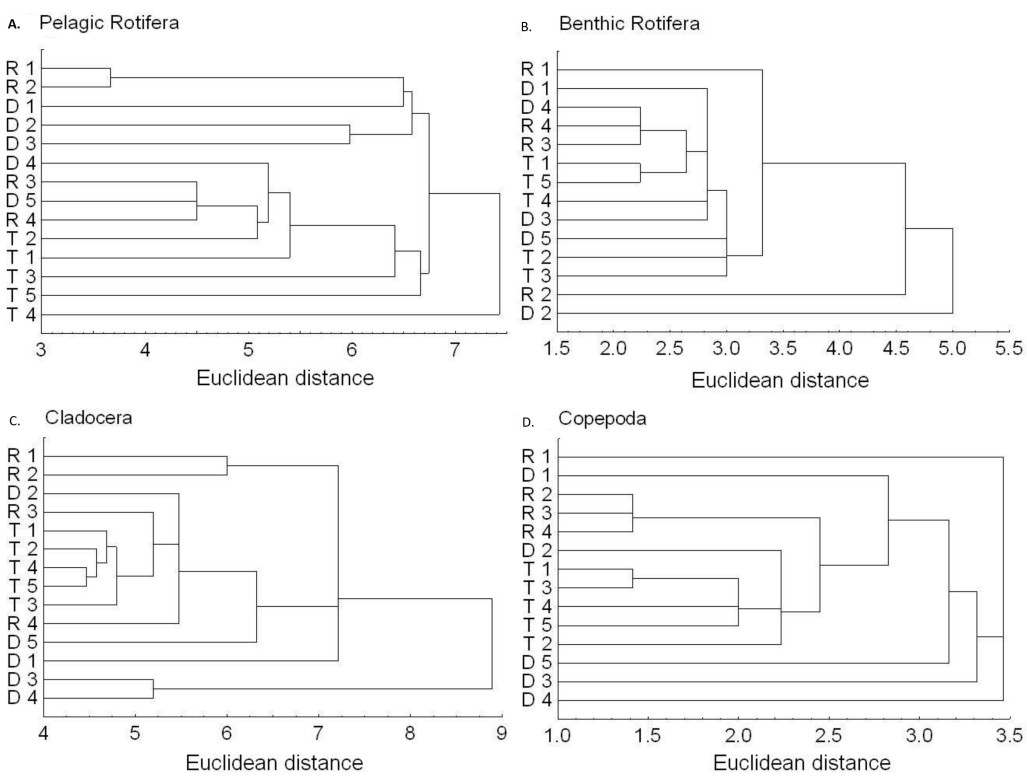

**Figure 4** **Dendrogram (cluster analysis) of the sites in the Barycz river.** R1–4, sites in the free-flowing waters of the Barycz river; D1–5, sites in the dams of the Barycz river; T1–5, sites in the tributaries of the Barycz river based on the richness of pelagic rotifers, benthic rotifers, cladocerans and copepods. (A) Pelagic Rotifera, (B) Benthic Rotifera, (C) Cladocera, (D) Copepoda.

**Table 4** **All significant differences (*P*-values) between the studied sites for the abundance of the zoo-plankton groups (post-hoc multiple comparisons of mean ranks for all groups).**

| Group | Site | T1 | D1 | R2 | T2 | D2 | T3 | T4 | T5 | R4 |
|---|---|---|---|---|---|---|---|---|---|---|
| Pelagic Rotifera | R1 | ** | | | *** | | *** | *** | *** | |
| Cladocera | R1 | * | | | *** | * | *** | *** | *** | |
| | T2 | | * | * | | | | | | |
| Nauplii | R1 | | | | ** | | * | | * | |
| | T2 | | | * | | | | | | |
| Copepoda | R1 | * | | | *** | ** | *** | *** | *** | |
| | T2 | | | | * | | | | | * |
| | T3 | | | | * | | | | | * |

**Notes.**
*$P < 0.05$.
**$P < 0.01$.
***$P < 0.001$.

sites was lower than in the tributaries, and higher than in the lotic sections. The abundance of cladocerans at R1 was significantly lower than at the tributary sites and D2 ($P < 0.05$) (Table 4). Copepods (all stages) obtained the lowest abundance at R1. The abundance of Nauplii at R1 was significantly lower than at T2, T3 and T5. While the abundance of other

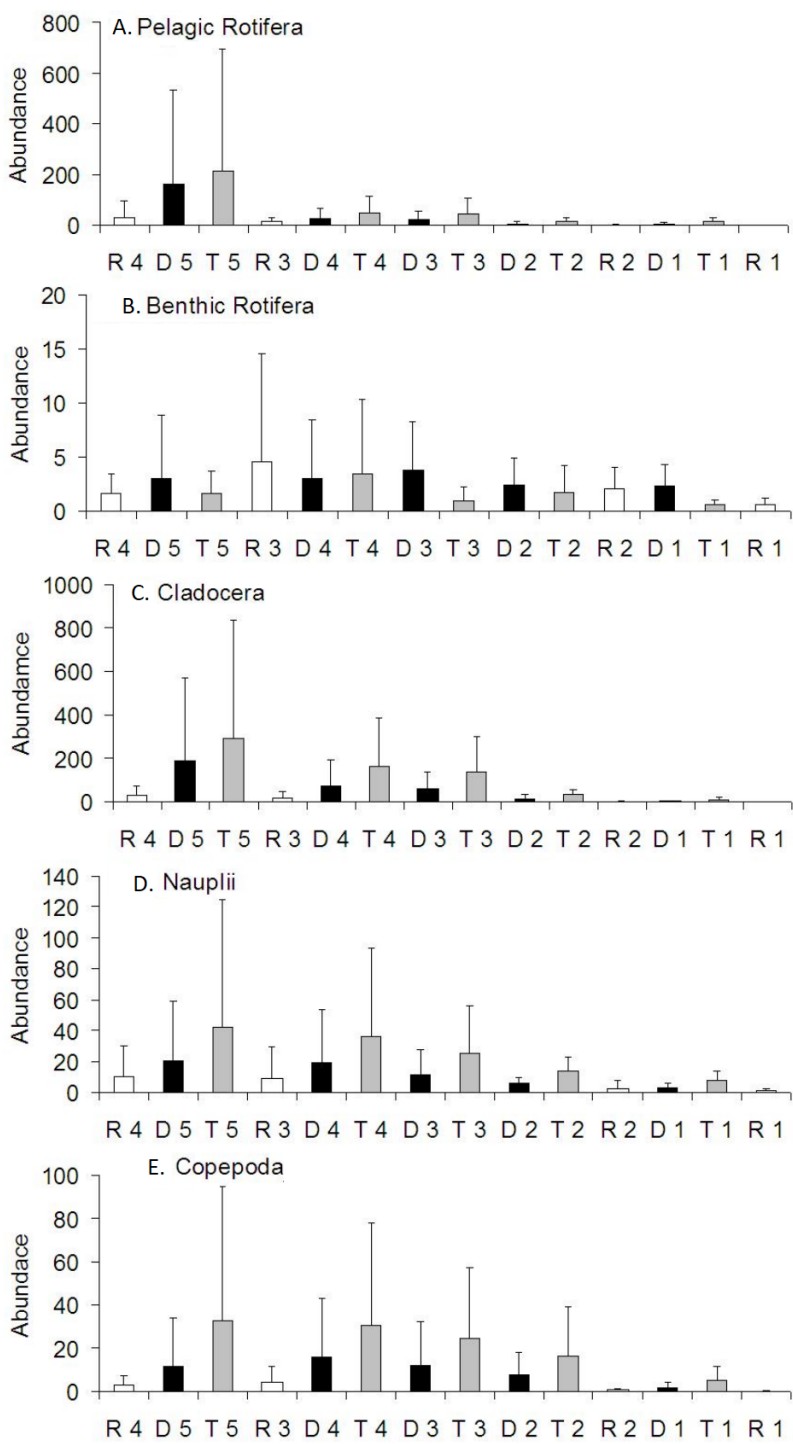

**Figure 5** **Spatial distribution of the zooplankton abundance in the Barycz river.** R1–4, sites in the free-flowing waters of the Barycz; D1–5, sites in the dams of Barycz; T1–5, sites in the tributaries of the Barycz. (A) Pelagic Rotifera, (B) Benthic Rotifera, (C) Cladocera, (D) Naupli, (E) Copepoda.

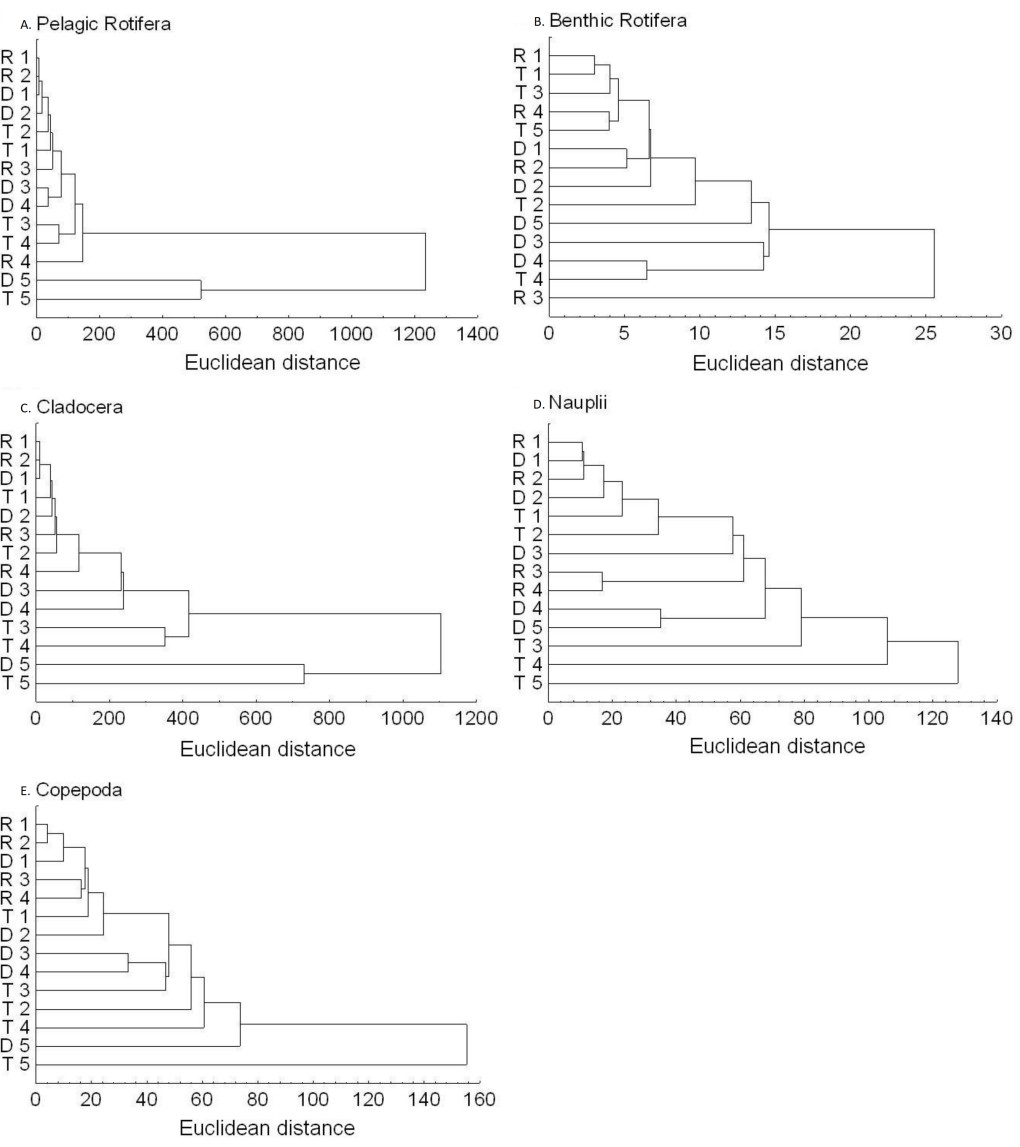

**Figure 6  Dendrogram (cluster analysis) of the sites in the Barycz river.** R1–4, sites in the free-flowing waters of the Barycz; D1–5, sites in the dams of the Barycz; T1–5, sites in the tributaries of the Barycz, based on the abundance of pelagic rotifers, benthic rotifers, cladocerans, Nauplii and copepods. (A) Pelagic Rotifera, (B) Benthic Rotifera, (C) Cladocera, (D) Naupli, (E) Copepoda.

copepods stages at R1 was significantly lower than at the sites of the tributaries ($P < 0.05$). Considering the subsequent sites below the tributaries, there were no significant differences in the abundance of crustacean groups between the sites ($P > 0.05$).

Based on the abundance of pelagic rotifers, cladocerans, copepods (Nauplii and the remaining stages), the cluster analysis shows a spatial similarity between the tributaries and the Barycz sites (Fig. 6). Such similarity is not observed when it comes to the abundance of benthic rotifers.

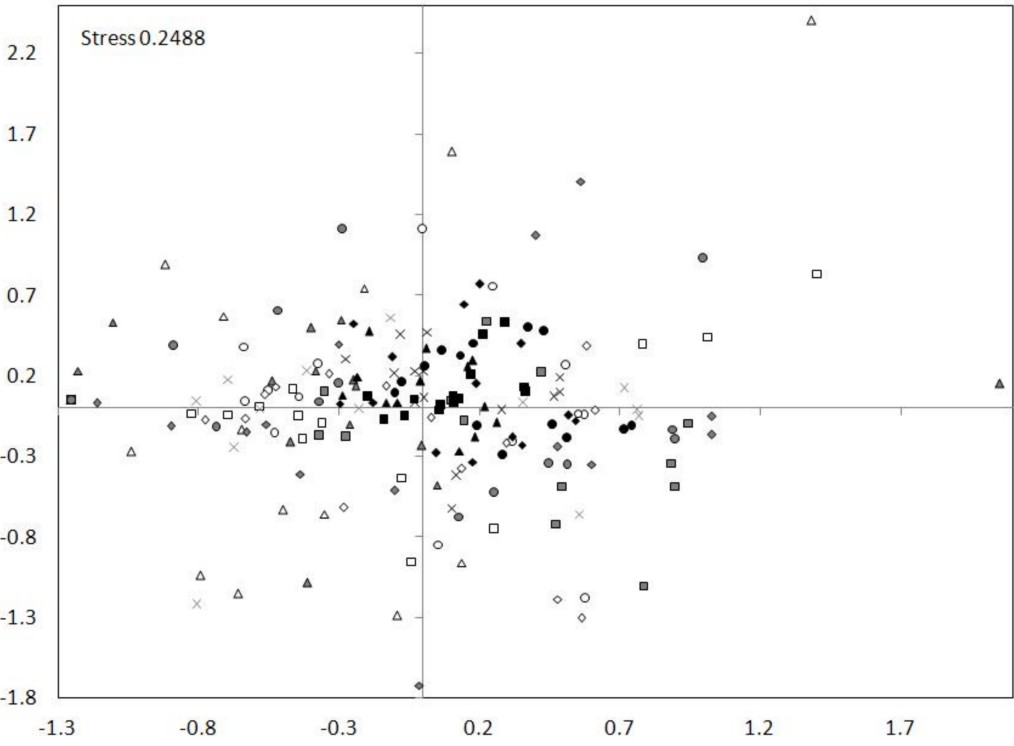

**Figure 7** **nMDS ordination for the abundance of all taxa at sites in the Barycz system.** The grouping in the nMDS ordination was based on the Bray–Curtis distances. Sites: white, lotic sections of the river; grey, dammed sections of the river; black, tributaries (carp pond outlets). White triangle, R1; white square, R2; white circle, R3; white rhombus, R4; grey triangle, D1; grey square, D2; grey circle, D3; grey rhombus, D4; grey cross (X), D5; black triangle, T1; black square, T2; black circle, T3; black rhombus, T4; black cross (X), T5.

The nMDS ordination showed a high similarity between the tributary sites. The dam sites exhibit less similarity and the lotic section sites showed the smallest similarity (Fig. 7). The nMDS ordination also indicated the greatest similarity in the zooplankton abundance occurred between the tributary sites and the dammed sites, and not between the tributary sites and the lotic section sites. The analysis showed also a similarity between subsequent sites (generally the distances between subsequent sites of tributary, dam and lotic section were the smallest).

At all sites, crustaceans were abundance dominants (Fig. 8). Between the tributaries and the sites at the dam or lotic sections located below, a similar percentage of the same groups comprised of the total zooplankton abundance (Fig. 8). At the lotic section sites (in relation with the dam upstream), a decrease in the percentage of abundance of cladocerans and copepods was observed. The same species were abundance dominants at the carp pond outlets (tributaries) and below their mouths in the Barycz (in the impoundments and lotic sections). This pattern was best manifested by *Bosmina longirostris, Chydorus sphaericus,* Nauplii Cyclopoida and *Acanthocyclops robustus* (Table 3).

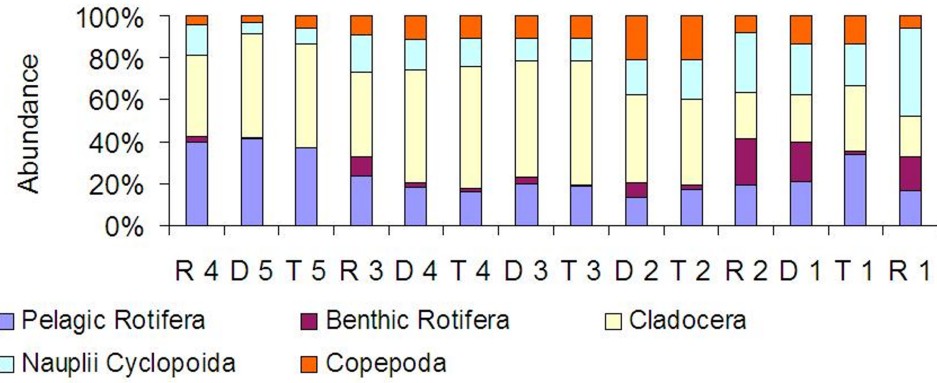

**Figure 8** **Percentage contribution of zooplankton groups in the mean total abundance of zooplankton at the sites examined in the Barycz river.**

**Table 5** **Spearman's significant correlations between richness and the values of environmental factors (P < 0.05).**

| Variable | Pelagic Rotifera | Benthic Rotifera | Cladocera | Copepoda |
|---|---|---|---|---|
| Chlor_a | 0.28 | 0.23 | 0.39 | 0.41 |
| Velocity | | | | −0.35 |
| ND | 0.32 | | 0.36 | |
| DCP | | | | −0.22 |
| NCO | 0.31 | | 0.35 | |
| TPelRotR | 0.49 | 0.31 | 0.55 | 0.47 |
| TBenRotR | | 0.31 | | |
| TClaR | 0.60 | | 0.62 | 0.36 |
| TCopR | 0.27 | 0.22 | 0.31 | 0.42 |

**Notes.**
Chlor_a, content of chlorophyll a; Velocity, current velocity; ND, number of dams above the site in the Barycz river; DCP, distance between the site in the Barycz river and the closest carp pond in the pond system; NCO, number of carp pond outlets above the site in the Barycz river; TPelRotR, richness of pelagic rotifers in the closest tributary above the site in the Barycz river; TBenRotR, richness of benthic rotifers in the closest tributary above the site in the Barycz river; TClaR, richness of cladocerans in the closest tributary above the site in the Barycz river; TCopR, richness of copepods in the closest tributary above the site in the Barycz river.

## Impact of environmental variables

At the Barycz sites, richness of each group positively correlated with the chlorophyll a content (P < 0.05) (Table 5). Pelagic rotifers and cladocerans richness also positively correlated with the values of ND and NCO (P < 0.05). Moreover, a negative correlation between the richness of copepods vs. the current velocity and the DCP was observed (P < 0.05). In the Barycz, richness of each group revealed a positive correlation with the richness of the same group in the closest tributary (carp pond outlet) (P < 0.05).

The abundance in the Barycz, a pattern of Spearman's significant correlation coefficients similar to that of richness was observed (Table 6). Each group correlated positively with the chlorophyll a content and negatively with the current velocity (P < 0.05). Abundance of copepods correlated negatively with the DCP values (P < 0.05). Abundance of each group

**Table 6** Spearman's significant correlations between abundance and the values of environmental factors ($P < 0.05$).

|  | Pelagic Rotifera | Benthic Rotifera | Cladocera | Nauplii Cyclopoida | Copepoda |
|---|---|---|---|---|---|
| Chlor_*a* | 0.32 | 0.30 | 0.50 | 0.45 | 0.40 |
| Velocity | −0.20 | −0.18 | −0.29 | −0.29 | −0.32 |
| ND | 0.33 |  | 0.28 |  |  |
| DCP |  |  |  |  | −0.26 |
| NCO | 0.34 |  | 0.28 |  |  |
| TPelRotA | 0.81 | 0.57 | 0.72 | 0.66 | 0.68 |
| TBenRotA |  | 0.39 | . | 0.29 |  |
| TClaA | 0.64 | 0.51 | 0.87 | 0.69 | 0.75 |
| TNaupCA | 0.60 | 0.56 | 0.76 | 0.82 | 0.76 |
| TCopA | 0.63 | 0.55 | 0.79 | 0.70 | 0.84 |

**Notes.**

Chlor_*a*, content of chlorophyll *a*; Velocity, current velocity; ND, number of dams above the site in the Barycz river; DCP, distance between the site in the Barycz river and the closest carp pond in the pond system; NCO, number of carp pond outlets above the site in the Barycz river; TPelRotA, abundance of the pelagic rotifers in the closest tributary above the site in the Barycz river; TBenRotA, abundance of benthic rotifers in the closest tributary above the site in the Barycz river; TClaA, abundance of cladocerans in the closest tributary above the site in the Barycz river; TNaupCA, abundance of copepods Nauplii in the closest tributary above the site in the Barycz river; TCopA, abundance of copepods in the closest tributary above the site in the Barycz river.

correlated positively with the abundance of the same group in the closest tributary (carp pond outlet) ($P < 0.05$).

## DISCUSSION

### Task 1. A comparison of the zooplankton communities between the free-flowing sections and impounded sections of the Barycz

Spatial pattern of zooplankton richness and abundance in the Barycz shows that the value of the zooplankton parameters increases with the distance from the river source. River Continuum Concept (*Vannote et al., 1980*) defines this as a phenomenon typically occurring in large rivers. However, in case of the Barycz continuum, it was the dam impoundments that affected the zooplankton communities as was seen by the increase in species numbers right after the dams. Generally, this phenomenon is discussed in the context of large rivers with large dams that create large reservoirs. In such large reservoirs richness and abundance of zooplankton is similar to that of lakes (*Akopian, Garnier & Pourriot, 1999*; *Doi et al., 2008*; *Pourriot, Rougier & Miquelis, 1997*; *Żurek & Dumnicka, 1989*). In regard with the present study results we could state that also small dams cause significant growth in zooplankton richness and abundance between the upstream and the downstream. A similar pattern of zooplankton community distribution in a small river impounded by human-made dams or beaver dams was observed by *Czerniawski (2013)*, *Czerniawski & Domagala (2014)* and *Zhou et al. (2008)*. We observed that in the impoundments of the Barycz, the zooplankton richness and abundance were high, and comparable to or even higher than those in typical temperate limnetic basins, e.g., eutrophic lakes or reservoirs (*Gołdyn & Kowalczewska-Madura, 2008*; *Karabin, Ejsmont-Karabin & Kornatowska, 1997*; *Lair, 2006*).

The main variables that contribute to increase the richness and abundance of zooplankton in the Barycz are: the decrease of the current velocity, longer water retention time or greater areas of open-water zones, a higher number of slackwater areas, floodplains and adjacent water bodies (*Richardson, 1992*; *Zhou et al., 2008*; *Nielsen et al., 2013*; *Czerniawski & Domagala, 2014*). The majority of microfauna, including freshwater zooplankton, is unable to persist if the current velocity is higher than $0.1 \text{ m s}^{-1}$ (*Lair, 2006*; *Richardson, 1992*). Such low current velocity values noted at the dam sites in the Barycz were favorable for the occurrence of zooplankton and for the increase of their populations.

The phenomenon that many species were observed in the Barycz but not in the pond outlets indicates that the dams provided advantageous conditions for the development of new species (not observed in upstream and in the carp pond outlets). This refers especially to small cladocerans (Alonidae) and littoral rotifers (*Euchlanis* sp., *Mytilina* sp.) occurring in an environment associated with macrophytes (*Rybak & Błędzki, 2010*). The more rapid increase in the richness of the abovementioned taxa, and the maintained high density of zooplankton in the impoundments was possible because features typical of stagnant water basins covered by macrophytes appeared in the impoundments (*Czerniawski, 2013*; *Czerniawski & Domagala, 2014*; *Radwan, 2004*; *Zhou et al., 2008*). Therefore, altered conditions in the impoundments led to the appearance of new species of pelagic, epiphytic and *epilithic,* rotifers and crustaceans, and, in turn, the growth of the zooplankton abundance (*Czerniawski, 2013*; *Czerniawski, Pilecka-Rapacz & Domagala, 2013*; *Nielsen et al., 2013*; *Richardson, 1992*). *Grabowska, Ejsmont-Karabin & Karpowicz (2013)* and *Czerniawski & Pilecka-Rapacz (2011)*; *Grabowska, Ejsmont-Karabin & Karpowicz (2013)* also observed a percentage domination in the abundance of small cladocerans in a shallow riverine section covered by macrophytes. *Richardson (1992)* and *De Bie et al. (2008)* found that the predominance of small cladocerans in running waters may be due to the fact that these species live in close association with the substratum or exhibit a strong habitat selection in favor of well-structured littoral zones, which may reduce their vulnerability to downstream washout. The bed covered with macrophytes is the key factor that positively affects the richness of zooplankton (*Kornijow et al., 2005*; *Kuczynska-Kippen & Nagengast, 2006*). In the impoundments of the Barycz, numerous slackwater areas were observed, from which zooplankton could be washed into the main channel (*Czerniawski, 2013*; *Nielsen, Gigney & Watson, 2010*; *Richardson, 1992*). Very good trophic conditions for filter-feeding plankters could have occurred in the Barycz dam impoundments because of the high concentration of chlorophyll *a*. Richness and abundance of each zooplankton group were significantly and positively correlated with the chlorophyll *a* values, which were also higher in the impoundments than in the lotic section. These effects can be due to the associated improved nutritional conditions for filter-feeding plankters. Such a correlation is more frequently observed in stagnant waters (*Gołdyn & Kowalczewska-Madura, 2008*; *Kamarainen et al., 2008*; *Levesque, Beisner & Peres-Neto, 2010*) and slow-flowing waters (*Czerniawski, 2012*; *Czerniawski, Slugocki & Kowalska-Goralska, 2016*). Despite low densities of *Euchlanis* sp., *Mytilina* sp. and Alonidae, they were good indicators of the changes in the zooplankton communities in the dammed Barycz river. So, not only macroinvertebrates or ichthyofauna react on the anthropogenic changes in river bed.

Perhaps Water Framework Directive (WFD) should take into account also the zooplankton to biological evaluation of rivers. However few years ago zooplankton was proposed as a good bioindicator of the lakes conditions. Zooplankton may play a pivotal role in the trophic network of stagnant basins ecosystems, but it has not been included in the WFD guidelines as part of biological evaluation, despite that many authors have shown strong indicative properties of zooplankton (*Jeppesen et al., 2011*; *Ejsmont-Karabin, 2012*; *Ejsmont-Karabin & Karabin, 2013*).

It is generally known that zooplankton disperse passively in aquatic ecosystems and they can colonize new habitats (*Havel & Shurin, 2004*). Also zooplankton transfer from lakes and reservoirs into the river is well documented (e.g., *Basu & Pick, 1997*; *Pourriot, Rougier & Miquelis, 1997*; *Czerniawski & Domagala, 2014*). Drifting zooplankton can colonize the river if its bed offer zones with low current velocity or with stagnant water e.g., slackwaters, impoundments, floodplains (*Czerniawski & Sługocki, 2017*; *Czerniawski & Sługocki, 2018*). The Barycz is a kind of impoundment–cascade system in which richness and abundance of zooplankton increase with the number of dams. The drift from the upstream impoundments would influence richness and abundance in the upstream and the dam sites. That is because what is present in a dam site, would first have to drift through an upstream site to get to a dam site. The dam sites provide a place for drifting zooplankton to proliferate, and increase in abundance, and as a result rare species (e.g., *Euchlanis* sp., *Mytilina* sp.), pelagic rotifers, and small cladocerans are more easily detected when sampling. Richness and abundance of pelagic rotifers and cladocerans correlated positively with the number of dams in river. Thus, passive zooplankton dispersion into the impoundments could be affected by the impoundments upstream, and mainly Rotifera, that demonstrate the best ability to colonize new habitats by short life span, reproduce through cyclical parthenogenesis, production the resting eggs that are easily transferred (*Ejsmont-Karabin & Kruk, 1998*; *Wallace, 2002*).

The zooplankton abundance was reduced along the section stretching from the dam sites to the lotic sections downstream. The decreased richness and density of zooplankton at the downstream sites typically occurs in the outlet sections of lakes or reservoirs and it depend on level of turbulence, values of discharge and current velocities. Higher values of these variables can have negative effect on fish predation on zooplankton, grazing by suspension-feeding, filter-feeding or net-spinning of macrozoobenthos, or settling to sediments (*Chang et al., 2008*; *Czerniawski, Pilecka-Rapacz & Domagala, 2013*; *Nielsen et al., 2013*; *Thorp & Casper, 2003*). In Barycz relatively low values of current velocity were observed. However, with the distance from the last dam, the zooplankton abundance and chlorophyll *a* concentration was rapidly reduced, so the impact of dam sites and carp ponds outlets on the zooplankton in lower section of Barycz and on their recipient—Oder, is rather small.

## Task 2. A comparison of the zooplankton communities between the carp pond outlets and the river

It is certain that because of the dam presence, richness and abundance of zooplankton in the Barycz were high, and they were higher than in the lotic sections of other rivers, and

even in the reservoirs of other larger rivers (*Akopian, Garnier & Pourriot, 1999*; *Doi et al., 2008*; *Pourriot, Rougier & Miquelis, 1997*). This pattern was affected by an additional factor, namely a connection between the river and the carp pond outlets which could have allowed the zooplankton to drift passively. We observed a similarity in zooplankton abundance between the carp pond outlets and the subsequent sites in the Barycz. This indicates that there is a influence of the carp pond outlets on the zooplankton abundance in the Barycz. While the differences between the carp pond outlets and the impoundments in the Barycz were particularly apparent when it comes to richness and species similarity.

It is surprising that crustaceans were the abundance dominants at all sites of the Barycz. The zooplankton in rivers, as well as small and large reservoirs, mainly consist of Cyclopoida nauplii and minor species of rotifers (*Akopian, Garnier & Pourriot, 1999*; *Czerniawski & Domagala, 2014*; *Doi et al., 2008*; *Pourriot, Rougier & Miquelis, 1997*; *Zhou et al., 2008*). In the present study, we have observed a different pattern, which was found in a non-typical river system. It was a carp pond outlet—impoundment—river system in which the composition of dominants depended on the sites that provided the best conditions for colonization and development. In this case, these were primarily carp ponds whose outlets carried zooplankton mass into the Barycz. The impoundments in which zooplankton from the tributaries remained and could maintain its abundance at a stable level played only a minor role. Dominants in the pond outlets that were carried into the Barycz were small cladocerans (bosminids and chydorids), that once passively drifted into the Barycz and dominated in the river as well. Bosminids and chydorids are small species that abundantly colonize both the pelagial and the littoral of either shallow and deep reservoirs, in which they can be dominant (*Gasiorowski & Szeroczynska, 2004*; *Rybak & Błędzki, 2010*; *Soto & De los Rios, 2006*).

*Cottenie et al. (2003)* claimed that connected water basins have similar zooplankton structures if they are environmentally similar. Results of our study show that the spatial pattern of zooplankton communities in the Barycz river confirms this statement. The number of taxa at environmentally similar sites (i.e., at different dam sites, different lotic sites and different tributary sites) was similar, although the species similarity between these sites was not high. Taxonomic similarity achieved high values between the subsequent sites, e.g., a tributary vs. an impoundment, or lotic section sites and dam sites vs. lotic section sites. Hence, the number of species was comparable at similar sites, however, these species were replaced by other species. This fact demonstrates the high influence of the tributaries and the dam sites on the development of zooplankton species structure in downstream locations.

To fully address the problem of carp pond influence on zooplankton population in the Barycz we would have to examine the zooplankton in the ponds and its similarity to the Barycz. However, it was not the objective of this study. We can be certain that the influx of zooplankton in the carp pond outlets would provide enough data to determine the influence of carp ponds on zooplankton and the Barycz. That is because plankters found in ponds dispersed passively to the outlets, which was proved by the similarities found between zooplankton communities in water basins and their outlet sections.

## CONCLUSION

Spatial changes in the zooplankton composition in the Barycz river reflected the effects of physical changes induced by the dam impoundments and by dispersion of zooplankton from the carp ponds outlets. In the case of pelagic rotifers, the higher the mean richness and abundance in the tributaries, the higher values of this parameters were observed at the subsequent sites below a tributary in the Barycz. Since the abundance of most groups (except for benthic rotifers) in the tributaries was similar to that at lower sites in the Barycz, the influence of the tributaries on the abundance in the dam and lotic sections is significant. Similarly significant was the influence of the impoundments on the zooplankton abundance in the lotic sections. The carp ponds outlets cause a large influx of zooplankton biomass to the river. Moreover, the river itself in its impounded sections provides advantageous conditions for retention and colonization by a high abundance of zooplankton dispersing from the carp ponds, and for the development of new species (not observed in the upstream), which, in turn, increases richness. This pattern was the most noticeable in the case of the crustaceans (cladocerans and copepods) which were characterized by a higher mean richness at the dam sites than in the tributaries. The Barycz is kind of an impoundment–cascade system, in which richness and abundance of zooplankton increase with the number of dams. However, richness and abundance decrease in case of the last site, despite the fact, that there is a number of dams below it and no pond influence. This shows that there is a clear influence of carp pond outlets, which are the source of zooplankton in the Barycz.

The presented study demonstrated how significant the effect of alterations to the river bed, as well as alterations to the water reservoirs in its catchment area (influx of the zooplankton from carp ponds) is to the biological changes in the river. The influx of live organic matter from the carp ponds and the dams has an impact on the zooplankton communities in the river downstream. The influx of zooplankton from carp ponds undoubtedly affected the increase in the zooplankton abundance in the river, while the dams had an effect on retaining this abundance and increasing richness.

### Funding
This work is supported by Wrocław Centre of Biotechnology, programme The Leading National Research Centre (KNOW) for years 2014–2018. The funders had no role in study design, data collection and analysis, decision to publish, or preparation of the manuscript.

### Grant Disclosures
The following grant information was disclosed by the authors:
Wrocław Centre of Biotechnology, programme The Leading National Research Centre (KNOW).

### Competing Interests
The authors declare there are no competing interests.

## Author Contributions

- Robert Czerniawski and Monika Kowalska-Góralska conceived and designed the experiments, performed the experiments, analyzed the data, contributed reagents/materials/analysis tools, prepared figures and/or tables, authored or reviewed drafts of the paper, approved the final draft.

## Data Availability

The raw data are provided as Supplemental Files.

## Supplemental Information

Supplemental information for this article can be found online at http://dx.doi.org/10.7717/peerj.5087#supplemental-information.

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
