# Peer review of "Spatial changes in zooplankton communities in a strong human-mediated river ecosystem"

_PeerJ, doi:10.7717/peerj.5087_

## Round 0.1 · original submission · Minor Revisions

The reviewers have highlighted the strengths of you submitted manuscript and have proposed a number of changes to improve the quality. Please consider the proposed changes as mandatory for a final acceptance of your manuscript.

In addition to the reviewer's comments I would like to draw your attention also to the following aspects: Keywords comprise "zooplankton" which is already part of the title and therefore should by replaced by another keyword. You have expressed your expectations in line 109ff. The introduction would benefit from a sharpened statement in the sense of a testable hypothesis which you could also incorporate into the abstract, together with a more detailed summary of results. I have checked your statistical analyses section and can confirm that the statistical aspect of your analysis is performed to the technical standard required for publication.

The clarity of figures was addressed by two reviewers with the proposal to order sampling sites in figures 3, 5 and 8 according to flow direction of the river from right to left.

Reviewers 2 and 3 have proposed a couple of linguistic improvements (cf. comments/changes in the annotated manuscripts) which you should carefully follow for the revision, including the avoidance of repetitions.

Reviewer 1 ·

Basic reporting

The manuscript has a clear structure which meets the requirements of an international journal. It is written in professional English, and the literature mentioned in the introduction is relevant and reflects the state of the art. This shows that the authors are experts in their field.
The figures and tables are relevant and are high quality, but some of them could be improved to make it easier to compare the sampling points with the flow direction of the river. Figure 3, Figure 5 and Figure 8 would be easier to understand if the order of the sampling stations were similar to the flow direction of the river Barycz, with RI, T1… starting on the right side of the graph.
Another comment concerns Table 2, where taxa, groups and, in the last line, the number of taxa are listed. It would be helpful to provide information regarding how the groups are related to the number of taxa. Maybe you could add a sentence in the table heading.
The raw data offered corresponded to the PeerJ policy, but please add the units.

Experimental design

The focus of the paper is to provide information about the development of metazooplankton in a river, based on river dams and the outlets of carp ponds. The findings are new, because previously published studies have only considered the influence of dams on the plankton community in large rivers, whereas the influence of ponds used for fish culture on the plankton community in rivers has not been considered until now. This means that this paper includes original primary research which is within the scope of the journal.
The standard of the investigation is appropriate, however, the manuscript would be improved by adding some more information about the sampling sites: 1) information about why you used these sampling points (and not others), 2) information about whether the water samples from the free-flowing section were collected through successive sampling using a Lagrangian sampling strategy and 3) information about the interaction between the carp ponds and the regulated channels. Perhaps an example outline would be helpful.
Do you have information about the water from the outlets (e.g. plankton, nutrients and seston)? I had the impression from your description that the water samples are “only” from the channels.

Validity of the findings

The findings are new (see the paragraph regarding the experimental design) and meaningful for understanding the riverine community.
There are two additional points which could be discussed:
1) What influence the observed development will have on the Oder River is one point for discussion. Do you think that the River Barycz is a promotor of the plankton densities in the Oder zooplankton community?
2) Your findings could also be interesting for the European Water Framework Directive.

The data were analysed with appropriate mathematical methods, and the latest literature is included.

I recommend the publication of this interesting and important paper. The expected changes are minor ones which could help to improve the clarity and the scope of the paper.

Reviewer 2 ·

Basic reporting

Abstract:
Should be rewriten. There is no one word on methods used, and results are not sufficiently described. If an aim of the paper was to examine the effects of carp ponds and dams, then what was a hypothesis? It should be given in Abstract and Introduction.
Table 2. Should be Brachionus calyciflorus and Scaridium longicaudum
Figures are not generally bad, but extremely difficult to understand if you have not enough time to spend on looking for explanations.
The manuscript is full of unnecessary introductory phrases as:
„regarding”, „it is worth”, „moreover”, „it must bepointed” etc.

The are also my remarks in the manuscript (pdf) attached

Experimental design

Well performed investigations sufficiently described in the manuscript

Validity of the findings

No comment

Additional comments

No comments

Annotated reviews are not available for download in order to protect the identity of reviewers who chose to remain anonymous.

Reviewer 3 ·

Basic reporting

There should be a more carefull use of English language. They way it is, the authors are trying to explain their point and repear the same thing in consecutive sentenses with different words. Proper language use will allow for this repeatition to be avoided, the text to become more concise and to the point.
Literature reference mostly OK
Structure of the article OK, raw data available
results relevant to hypotheses

Experimental design

The manuscript deals with a very interesting subject, within the Scope of the Journal, the effect of the dams on species distribution, especially zooplankton which lately understudied in freshwater ecosystems due to the fact that it is excluded by the WFD 2000/60. Also a very strong asset of this study is the large data set.
They define the research questions and apply standard methods sufficiently described

Validity of the findings

As I already stated, a very strong asset of this study is the large data set.

But in the wordy text the authors fail to link their results with the conclusions

Additional comments

My specific comments can be found directly on the PDF

Annotated reviews are not available for download in order to protect the identity of reviewers who chose to remain anonymous.

---

## Round 0.2 · accepted · Accept

Thank you very much for the thorough revision of the manuscript and the consideration of the reviewers' comments.

#